# Partial verification bias correction using scaled inverse probability resampling for binary diagnostic tests

**Wan Nor Arifin**[1]*, **Umi Kalsom Yusof**[2]*

**1** Biostatistics and Research Methodology Unit, School of Medical Sciences, Universiti Sains Malaysia, Kelantan, Malaysia, **2** Albukhary International University, Alor Setar, Kedah, Malaysia

* wnarifin@usm.my (WNA); umi.yusof@aiu.edu.my (UKY)

**Data availability statement:** The code and datasets used in this study are available at the following GitHub repository: https://github.com/wnarifin/sipw_in_pvb.

## Abstract

Diagnostic accuracy studies are crucial for evaluating new tests before their clinical application. These tests are compared with gold standard tests, and accuracy measures such as sensitivity (Sn) and specificity (Sp) are often calculated. However, these studies frequently suffer from partial verification bias (PVB) due to selective verification of patients, which leads to biased accuracy estimates. Among the methods for correcting PVB under the missing-at-random assumption for binary diagnostic tests, a bootstrap-based method known as the inverse probability bootstrap (IPB) was proposed. IPB demonstrated low bias for estimating Sn and Sp, but exhibited higher standard errors (SE) than other PVB correction methods and only corrected the distribution of the verified portion of the PVB data. This paper introduces two new methods to address these limitations: scaled inverse probability weighted resampling (SIPW) and scaled inverse probability weighted balanced resampling (SIPW-B), both built upon IPB. Using simulated and clinical datasets, SIPW and SIPW-B were compared against IPB and other existing methods (Begg and Greenes', inverse probability weighting estimator, and multiple imputation). For the simulated data sets, different combinations of disease prevalence (0.4 and 0.1), Sn (0.3 to 0.9), Sp (0.6 and 0.9), and sample sizes (200 and 1000) were generated. Two commonly used clinical datasets in PVB correction studies were also used. Performance was evaluated using bias and SE for the simulated data. Simulation results showed that both new methods outperformed IPB by producing lower bias and SE for Sn and Sp estimation, showing results comparable to existing methods, and demonstrating good performance at low disease prevalence. In clinical datasets, SIPW and SIPW-B were consistent with existing methods. The new methods also improve upon IPB by allowing full data restoration. Although the methods are computationally demanding at present, this limitation is expected to become less important as computing power continues to increase.

## Introduction

Diagnostic tests are crucial in medical care, so ensuring their clinical validity through diagnostic accuracy studies is essential [1,2]. These studies involve comparing a new test with an

**Funding:** Funding 1: WNA: This is generic funding for publication without grant number. Research Creativity and Management Office, Universiti Sains Malaysia https://research.usm.my/. The funder did not have any role in the study design, data collection and analysis, decision to publish, or preparation of the manuscript. Funding 2: WNA: This is generic funding for publication without grant number. School of Medical Sciences, Universiti Sains Malaysia. https://rni.kk.usm.my/. The funder did not have any role in the study design, data collection and analysis, decision to publish, or preparation of the manuscript.

**Competing interests:** The authors have declared that no competing interests exist.

established gold standard to evaluate its performance using accuracy measures such as sensitivity (Sn) and specificity (Sp) for binary tests [1,3–5]. However, verifying disease status with the gold standard can be expensive, time-consuming, and invasive [1,5–8]. This verification challenge often leads to partial verification bias (PVB), where patients with positive test results are predominantly selected for gold standard verification, while fewer patients with negative test results are verified [1,6,8,9]. This gives rise to a missing-at-random (MAR) missing data mechanism, as the decision to verify depends on the diagnostic test result [5,6].

It is important to correct bias during analysis because PVB leads to biased estimates of diagnostic accuracy measures [1,6,10]. This can affect clinical practice, as biased estimates may result in clinically invalid tests and incorrect clinical decisions [2,6]. Methods for correcting PVB have been comprehensively reviewed elsewhere [2,11]. For binary tests under the MAR assumption, PVB correction methods can be categorized into Begg and Greenes' (BG)-based, propensity score (PS)-based, and multiple imputation (MI) methods. BG-based and MI methods adjust accuracy measures by calculating the probability of disease status given the test result, which are unbiased under MAR assumptions [3,12]. In contrast, PS-based methods estimate the probability of verification given the test result and use weighting to correct the bias [13,14]. Several medical studies have applied these correction methods in their research, highlighting the importance of the bias correction [15–19].

Inverse probability bootstrap (IPB) sampling was introduced to address sampling bias in model-based analyses [20]. While the bootstrap technique is traditionally used for estimating standard errors (SE), IPB leverages this technique to achieve unbiased parameter estimates by creating weighted samples. This method corrects the sample distribution without requiring extensive modifications or new methods [20]. Because it is a bootstrap approach, IPB simplifies the estimation of SE and enables the calculation of confidence intervals for statistical inference [20].

The PS-based method for PVB correction and IPB share a common approach; both begin by estimating the selection probability, or verification probability in the context of PVB, and then use this probability to correct bias through weighting methods. The IPB approach offers an attractive way to address bias because it relies on the bootstrap technique, which provides several advantages. Therefore, IPB was adapted in the context of PVB correction in a study by Arifin and Yusof [21].

In their study [21], IPB demonstrated low bias in the estimation of Sn and Sp. However, it exhibited a relatively higher SE than other PVB correction methods and only corrected the distribution of the verified portion of the PVB data. To address these limitations, this study proposes two methods: scaled inverse probability weighted resampling (SIPW) and scaled inverse probability weighted balanced resampling (SIPW-B), designed for PVB correction under the MAR assumption for binary diagnostic tests.

## Materials and methods

The simulated and clinical data sets used in this study, the proposed methods for PVB correction based on IPB, the metrics for performance evaluation, the selected methods for comparison, and the experimental setup are described in this section. The following notations are used: $T$ = test result, $D$ = disease status, $V$ = verification status, $n_1$ = verified observations, $n_0$ = unverified observations, and $n$ = all observations, or $n_1 + n_0$.

## Data sets

This study evaluated and compared different methods using both simulated and clinical datasets. Simulated data allowed performance assessment against known parameter values [20,22], while clinical data enabled comparisons with established reference data, following the practice of previous PVB correction studies [23–26].

**Simulated data sets.** The simulated data sets were generated using the settings described in [21], which were adapted from [24–26]. The settings are outlined as follows:

1. True disease prevalence ($p$) or $P(D = 1)$: moderate = 0.40 and low = 0.10.
2. True sensitivity (Sn) $P(T = 1|D = 1)$: low = 0.3, moderate = 0.6, high = 0.9
3. True specificity (Sp) $P(T = 0|D = 0)$: moderate = 0.6, high = 0.9.
4. Verification probabilities: When verification depends only on the test result, this represents an MAR missingness mechanism. Fixed verification probabilities given the test result $P(V = 1|T = t)$, were set at $P(V = 1|T = 1) = 0.8$ and $P(V = 1|T = 0) = 0.4$ [24]. In other words, patients with positive test results are more likely to be verified, with a probability of 0.8. Conversely, patients with negative test results are less likely to be verified, with a probability of 0.4.
5. Sample sizes, $n$: 200 and 1000.

For the complete data, the probabilities of the counts in a $2 \times 2$ cross-tabulated table of $T$ versus $D$ follow a multinomial distribution [24,26]. Based on pre-specified values of Sn = $P(T = 1|D = 1)$, Sp = $P(T = 0|D = 0)$ and $p = P(D = 1) = \pi$, the probabilities of the counts are given by $M(\pi_1, \pi_2, \pi_3, \pi_4)$, where

$$\pi_1 = P(T = 1, D = 1) = P(T = 1|D = 1)P(D = 1),$$
$$\pi_2 = P(T = 0, D = 1) = P(T = 0|D = 1)P(D = 1) = [1 - P(T = 1|D = 1)]P(D = 1),$$
$$\pi_3 = P(T = 1, D = 0) = P(T = 1|D = 0)P(D = 0) = [1 - P(T = 0|D = 0)]P(D = 0),$$
$$\pi_4 = P(T = 0, D = 0) = P(T = 0|D = 0)P(D = 0).$$

For each specified sample size $n$, generating a simulated data set with MAR-induced PVB involves the following steps:

1. A complete data set of size $n$, following a multinomial distribution, $M(\pi_1, \pi_2, \pi_3, \pi_4)$ was generated. Numerical values were randomly drawn from 1, 2, 3, 4 according to these probabilities.
2. The values were converted into realizations of the $T = t$ and $D = d$ variables, where the numbers were mapped as follows: $1 \rightarrow (T = 1, D = 1)$, $2 \rightarrow (T = 0, D = 1)$, $3 \rightarrow (T = 1, D = 0)$ and $4 \rightarrow (T = 0, D = 0)$.
3. Under the MAR assumption, the PVB data set was generated by adding $V = \{1, 0\}$ with verification probabilities of $P(V = 1|T = 1) = 0.8$ and $P(V = 1|T = 0) = 0.4$, where $V$ follows a binomial distribution. $D$ values for $V = 0$ observations were set to $NA$ to create missing values.

The layout of the simulated data for both the complete and PVB data sets is shown in Fig 1.

**Clinical data sets.** This study utilized two commonly used clinical data sets [24,26–30] to show and compare the implementation of the PVB correction methods in real-world settings. The original data from these studies were converted to an analysis-ready format (.csv). These data sets are described as follows:

**Fig 1. Simulated data layout for complete and PVB data.**

1. **Hepatic scintigraphy test**

   The data set relates to hepatic scintigraphy, a diagnostic imaging technique for detecting liver cancer, as reported in the original study [27]. The test was performed on 650 patients, with 344 patients verified by liver pathological examination (gold standard). The percentage of unverified patients was 47.1%. The data set includes the following variables:

   - **Liver cancer**, *disease*: Binary, 1 = Yes, 0 = No
   - **Hepatic scintigraphy**, *test*: Binary, 1 = Positive, 0 = Negative
   - **Verified**, *verified*: Binary, 1 = Yes, 0 = No

2. **Diaphanography test**

   The data set relates to the diaphanography test for detecting breast cancer, as reported in the original study [28]. Diaphanography is a noninvasive breast examination method that uses transillumination with visible or infrared light to detect breast cancer. The test was performed on 900 patients, but only 88 patients were verified by breast tissue biopsy for histological examination (gold standard). The percentage of unverified patients was 90.2%. The data set includes the following variables:

   - **Breast cancer**, *disease*: Binary, 1 = Yes, 0 = No
   - **Diaphanography**, *test*: Binary, 1 = Positive, 0 = Negative
   - **Verified**, *verified*: Binary, 1 = Yes, 0 = No

Cross-tabulations of these data sets are given in Fig 2.

## Proposed methods

**Scaled inverse probability weighted resampling.** Based on the IPB method proposed by Arifin and Yusof [21], the scaled inverse probability weighted resampling (SIPW) method is proposed to overcome the limitations of IPB in correcting PVB. The SIPW algorithm is shown in Algorithm 1. IPB defines $n$ as the observed sample size, while SIPW defines $n$ as the complete sample size, including both verified and unverified samples. The verified sample size is $n_1$, and the unverified sample size is $n_0$. A sample of size $n$ is drawn with replacement $b$ times from $n_1$, producing $b$ samples. Since SIPW does not perform sampling with replacement to match the original $n_1$ size, these are not bootstrap samples as implemented in IPB. Therefore, the general term *resampling* is used in the method name.

**Scaled inverse probability weighted balanced resampling.** During this research, it was observed that subgroup sizes, $n_{D=1}$ and $n_{D=0}$ for diseased and non-diseased observations respectively, directly affect the precision (i.e. the SE) of Sn or Sp, as smaller sample sizes result in lower precision (higher SE). The SE, especially for Sn, can increase further when disease

**Algorithm 1 Scaled inverse probability weighted resampling.**

**Data, notations and definitions:**

Test status $T$, where $T = \{\text{Positive: 1, Negative: 0}\}$
Disease status $D$, where $D = \{\text{Yes: 1, No: 0, Unknown: NA}\}$
Verification status $V$, where $V = \{\text{Yes: 1, No: 0}\}$
PVB data $Y = \{T, D, V\}$ of size $n \times 3$
Individual patient instance is denoted with subscript $i = 1, \ldots, n$
$D$ consists of verified ($V = 1$) and unverified ($V = 0$) sets, where $D = \{D^1, D^0\}$
Unverified set of data $Y^0$ is of size $n_0$, consisting of instances with $V_i = 0$
Verified set of data $Y^1$ is of size $n_1 = n - n_0$, consisting of instances with $V_i = 1$
Propensity score for $i$th patient $PS_i$ is $\hat{P}(V_i = 1 | T_i)$
Inverse probability weight for $i$th patient $IPW_i$ is $\frac{V_i}{PS_i} + \frac{1 - V_i}{1 - PS_i}$
Scaled inverse probability weight for $i$th patient $SIPW_i$ is $\frac{IPW_i}{\sum_{i=1}^{n_1} IPW_i}$, such that $\sum_{i=1}^{n_1} SIPW_i = 1$
Sensitivity $Sn$ is $P(T = 1 | D = 1)$
Specificity $Sp$ is $P(T = 0 | D = 0)$

**Procedure:**

1: Estimate $\widehat{PS}_i$, using $Y$ by a logistic regression
2: Calculate $\widehat{IPW}_i \leftarrow \frac{1}{\widehat{PS}_i}$ for instances with $V_i = 1$
3: Calculate $\widehat{SIPW}_i \leftarrow \frac{\widehat{IPW}_i}{\sum_{i=1}^{n_1} \widehat{IPW}_i}$ for instances with $V_i = 1$
   /*—Begin resampling block—*/
4: $Y^{List} \leftarrow$ Empty list of size $b$ samples
5: $i \leftarrow 1$
6: **while** $i < b + 1$ **do**
7: $Y_i \leftarrow$ Sample $n$ instances from $Y^1$ of $n_1$ instances with probability $SIPW_i$
   /*Ensure valid $Y_i$, where $T \times D$ cross-classification table dimension $TblDim$ is 4*/
8: **if** $TblDim < 4$ **then**
9: Discard $Y_i$
10: Repeat sampling, $i \leftarrow i$
11: **else if** $TblDim = 4$ **then**
12: Save $i$th sample, $Y_i^{List} \leftarrow Y_i$
13: Continue with next $i$, $i \leftarrow i + 1$
14: **end if**
15: **end while**
   /*—End resampling block—*/
16: $Sn^{List} \leftarrow$ Empty list of size $b$ samples
17: $Sp^{List} \leftarrow$ Empty list of size $b$ samples
18: **for** $i = 1$ **to** $b$ in $Y^{List}$ **do**
19: $Sn_i^{List} \leftarrow$ Estimate $\widehat{Sn}_i$ from $T \times D$ cross-classification table
20: $Sp_i^{List} \leftarrow$ Estimate $\widehat{Sp}_i$ from $T \times D$ cross-classification table
21: **end for**
22: Estimate $\widehat{Sn} \leftarrow \frac{1}{b} \sum_{i=1}^{b} \widehat{Sn}_i$ from $Sn^{List}$
23: Estimate $\widehat{Sp} \leftarrow \frac{1}{b} \sum_{i=1}^{b} \widehat{Sn}_i$ from $Sp^{List}$

Hepatic scintigraphy test data

| Test | Liver cancer status | | | |
|---|---|---|---|---|
| | Cancer | No cancer | Unverified | Total |
| Positive | 231 | 32 | 166 | 429 |
| Negative | 27 | 54 | 140 | 221 |
| Total | 258 | 86 | 306 | 650 |

Diaphanography test data

| Test | Breast cancer status | | | |
|---|---|---|---|---|
| | Cancer | No cancer | Unverified | Total |
| Positive | 26 | 11 | 30 | 67 |
| Negative | 7 | 44 | 782 | 833 |
| Total | 33 | 55 | 812 | 900 |

**Fig 2. Cross-tabulation of hepatic scintigraphy and diaphanography data sets.**

prevalence decreases because the subgroup size of $n_{D=1}$ also becomes smaller. To reduce the SE, a possible solution was inspired by the way diagnostic accuracy studies are designed. A diagnostic accuracy study is ideally designed as a cohort study [3,4,31]. When the disease is very rare, a case-control study is more practical, where patients with the disease (cases) are compared to those without the disease (controls), often using a 1:1 to 1:3 ratio [31,32]. This approach is similar to data-level resampling methods used to address class imbalance in machine learning, where class distribution is adjusted through over- or under-sampling methods [33].

Therefore, the scaled inverse probability weighted balanced resampling (SIPW-B) method is proposed to mimic the case-control study design. SIPW-B balances subgroup sizes by resizing the sizes to a predefined ratio of $n_{D=0}:n_{D=1}$, while keeping the original sample size $n = n_{D=0} + n_{D=1}$. The SIPW-B algorithm is shown in Algorithm 2. In the algorithm, the target control:case ratio or $n_{D=0} : n_{D=1}$ is achieved through the following steps:

1. Set the desired relative size or ratio of control:case.
2. Calculate the initial relative size of $n_{D=0}$ to $n_{D=1}$ in the PVB sample.
3. Update $IPW_i$ for instances with $D_i = 1$ by multiplying the value with the initial relative size.
4. Calculate $k$ as the sum of $IPW_i$ for instances with $D_i = 0$, divided by sum of $IPW_i$ for instances with $D_i = 1$.
5. Update $IPW_i$ for instances with $D_i = 1$ by multiplying the value with $\frac{k}{\text{Desired relative size}}$
6. Calculate $SIPW_i$ as $IPW_i$ divided by the sum of $IPW_i$.

For the experiments in this study, the target relative size of control:case was set at 1, or a ratio 1:1.

**Algorithm 2 Scaled inverse probability weighted balanced resampling.**

**Data, notations and definitions:**

Test status $T$, where $T = \{\text{Positive: } 1, \text{ Negative: } 0\}$
Disease status $D$, where $D = \{\text{Yes: } 1, \text{ No: } 0, \text{ Unknown: } NA\}$
Verification status $V$, where $V = \{\text{Yes: } 1, \text{ No: } 0\}$
PVB data $Y = \{T, D, V\}$ of size $n \times 3$
Individual patient instance is denoted with subscript $i = 1, ..., n$
$D$ consists of verified ($V = 1$) and unverified ($V = 0$) sets, where $D = \{D^1, D^0\}$
Unverified set of data $Y^0$ is of size $n_0$, consisting of instances with $V_i = 0$
Verified set of data $Y^1$ is of size $n_0 = n - n_0$, consisting of instances with $V_i = 1$
Propensity score for $i$th patient $PS_i$ is $\hat{P}(V_i = 1 | T_i)$
Inverse probability weight for $i$th patient $IPW_i$ is $\frac{V_i}{PS_i} + \frac{1 - V_i}{1 - PS_i}$
Desired relative size of $n_{D=0} : n_{D=1}$, $RelSize$ i.e. control:case size
Scaled inverse probability weight for $i$th patient $SIPW_i$ is $\frac{IPW_i}{\sum_{i=1}^{n_{D=1}} IPW_i}$, such that $\sum_{i=1}^{n_{D=1}} SIPW_i = 1$
Sensitivity $Sn$ is $P(T = 1 | D = 1)$ and Specificity $Sp$ is $P(T = 0 | D = 0)$

**Procedure:**

1: Estimate $\widehat{PS}_i$, using $Y$ by a logistic regression
2: Calculate $\widehat{IPW}_i \leftarrow \frac{1}{\widehat{PS}_i}$ for instances with $V_i = 1$
   /*Balance $D = 1$ size relative to $D = 0$ size*/
3: Calculate $RelSizeInit \leftarrow \frac{n_{D=0}}{n_{D=1}}$
4: Update $\widehat{IPW}_i \leftarrow \widehat{IPW}_i \times RelSizeInit$ for instances with $D_i = 1 \cap V_i = 1$
5: Calculate constant $k \leftarrow \sum_{i=1}^{n_{D=0}} \widehat{IPW}_i / \sum_{i=1}^{n_{D=1}} \widehat{IPW}_i$
6: Update $\widehat{IPW}_i \leftarrow \widehat{IPW}_i \times (k / RelSize)$ for instances with $D_i = 1 \cap V_i = 1$
7: Calculate $\widehat{SIPW}_i \leftarrow \widehat{IPW}_i / \sum_{i=1}^{n_{D=1}} \widehat{IPW}_i$ for instances with $V_i = 1$
   /*—Begin resampling block—*/
8: $Y^{List} \leftarrow$ Empty list of size $b$ samples
9: $i \leftarrow 1$
10: **while** $i < b + 1$ **do**
11:    $Y_i \leftarrow$ Sample $n$ instances from $Y^1$ of $n_{V=1}$ instances with probability $SIPW_i$
   /*Ensure valid $Y_i$, where $T \times D$ cross-classification table dimension $TblDim$ is 4*/
12:    **if** $TblDim < 4$ **then**
13:        Discard $Y_i$
14:        Repeat sampling, $i \leftarrow i$
15:    **else if** $TblDim = 4$ **then**
16:        Save $i$th sample, $Y_i^{List} \leftarrow Y_i$
17:        Continue with next $i$, $i \leftarrow i + 1$
18:    **end if**
19: **end while**
   /*—End resampling block—*/
20: $Sn^{List} \leftarrow$ Empty list of size $b$ samples
21: $Sp^{List} \leftarrow$ Empty list of size $b$ samples
22: **for** $i = 1$ **to** $b$ in $Y^{List}$ **do**
23:    $Sn_i^{List} \leftarrow$ Estimate $\widehat{Sn}_i$ from $T \times D$ cross-classification table
24:    $Sp_i^{List} \leftarrow$ Estimate $\widehat{Sp}_i$ from $T \times D$ cross-classification table
25: **end for**
26: Estimate $\widehat{Sn} \leftarrow \frac{1}{b} \sum_{i=1}^{b} \widehat{Sn}_i$ from $Sn^{List}$
27: Estimate $\widehat{Sp} \leftarrow \frac{1}{b} \sum_{i=1}^{b} \widehat{Sn}_i$ from $Sp^{List}$

## Performance evaluation

The performance evaluation used metrics that measure the difference between an estimate and its true value [22,34,35]. For a finite number of simulations $B$, the selected metrics are calculated as follows:

1. **Bias**
   Bias of a point estimator $\hat{\theta}$ is defined as the difference between its expected value and the true value of a parameter $\theta$ [35]. It is calculated as:

$$Bias = E[\hat{\theta}] - \theta = \frac{1}{B}\sum_{i=1}^{B}\hat{\theta}_i - \theta. \tag{1}$$

2. **Standard error**
   Standard error (SE) is the square root of the variance and is calculated as:

$$SE = \sqrt{Var(\hat{\theta})} = \sqrt{\frac{1}{B-1}\sum_{i=1}^{B}(\hat{\theta}_i - \bar{\theta})^2}, \tag{2}$$

where $\bar{\theta}$ is the mean of $\hat{\theta}_i$ across repetitions.

Bias is often the primary metric of interest [22] because it reflects the accuracy of a method [35] and whether, on average, the method targets the parameter $\theta$ [22]. SE reflects the precision of the method [22,35], with a smaller SE indicating higher precision [35].

## Methods for comparison

The proposed SIPW and SIPW-B were compared with selected existing PVB correction methods, which are the BG method (representing BG-based methods), the inverse probability weighting estimator (IPWE) and IPB methods (representing PS-based methods), and the MI method (representing the imputation-based methods). These methods were chosen to represent different approaches for PVB correction. In addition, two more methods were included for comparison: full data analysis (FDA), serving as an ideal benchmark when complete and unbiased data are available [3]; and complete case analysis (CCA), an uncorrected method that exhibits bias in the presence of PVB [36]. Details of Sn and Sp calculations for these existing methods have been described elsewhere [11,21].

For the simulated datasets, the methods were compared based on the mean of the estimates, bias and SE, organized by sample size and Sn-Sp combination. Coverage, defined as the proportion of times the confidence interval (CI) includes the true estimate [22,34,35], was not included as a performance metric in our simulation because we did not propose a new method for obtaining the CI for SIPW and SIPW-B. For the clinical data sets, point estimates and their 95% CIs were compared.

## Experimental setup

All experiments were conducted using R version 3.6.3 [37] within the RStudio integrated development environment [38]. The final stable version of the 3.x.x series was chosen to ensure reproducibility, as the 4.x.x series is still under active development. The *mice* [39] (version 3.14.0) and *simstudy* [40] (version 0.5.0) R packages were used. A random seed of 3209673 was set for the entire study.

**Simulation setup and data analysis.** To test the performance of PVB correction methods, simulated data sets were generated following the procedures described in the *Simulated data sets* subsection above. This step was followed by analysis using FDA, CCA, BG, IPWE,

MI, IPB, SIPW, and SIPW-B. The general settings for data generation and analysis were: number of simulation runs $B = 500$, samples $b = 1000$ (for IPB, SIPW, and SIPW-B), and imputations $m = 100$ [41,42]. There were 12 different combinations of experimental settings (disease prevalence, true Sn, true Sp) at two sample sizes ($n = 200, 1000$). The general steps for data generation and analysis were as follows:

1. For a selected combination of experimental settings (for example, $p = 0.4$, Sn = 0.3, Sp = 0.6), complete data sets were generated, followed by MAR-induced PVB data sets.
2. Each generated data set was checked for validity, where it had to form a $2 \times 2$ cross-tabulated table ($T = 1, 0$ versus $D = 1, 0$) with no zero cell count. Any invalid data set was discarded.
3. The generation process continued until $B = 500$ valid complete and PVB data sets were obtained for each sample size ($n = 200$ and $n = 1000$), resulting in four sets of data sets in total.
4. Sn and Sp were estimated for each data set. Complete data sets were analyzed using the FDA method, while PVB data sets were analyzed by CCA, BG, IPWE, MI, IPB, SIPW, and SIPW-B. Mean estimates, bias, and SE were then calculated.
5. The steps above were repeated for each combination of experimental settings for all 12 combinations.

The code to generate and analyze the simulated data sets is available at https://github.com/wnarifin/sipw_in_pvb.

**Clinical data analysis.** Clinical data sets were analyzed using CCA, BG, IPWE, MI, IPB, SIPW and SIPW-B. For CCA, CIs for Sn and Sp were calculated using Wald interval, while for the BG method, the calculation steps from the original article were followed [43]. For IPWE, IPB, SIPW and SIPW-B, CIs were obtained by bootstrap percentile interval method [44]. For MI, CIs were obtained using Rubin's rule [11,45]. IPB did not require an additional bootstrapping step to obtain its CI because it already generated valid bootstrap samples for CI estimation [20,21]. Analysis settings were: number of samples $b = 1000$ (for IPB, SIPW and SIPW-B), bootstrap replicates $R = 1000$ (to obtain the CIs for IPWE, SIPW and SIPW-B), and $m =$ the percentage of incomplete cases for real clinical data sets [46–48]. The clinical data sets and code to reproduce the results are available at https://github.com/wnarifin/sipw_in_pvb.

## Ethics

The simulation component of this study did not involve human participants or any identifiable personal data. The simulated data were generated using the computational settings described previously. The clinical data analysis component used publicly available, aggregated secondary data to apply the selected methods. Therefore, no ethical approval was needed for either component of this research.

## Results

### Simulated data analysis

The simulation results for FDA, CCA and the PVB correction methods for $p = 0.4$ are displayed in Table 1, comparing sample sizes $n = 200$ and $1000$. The results are organized by parameter combinations of Sn = (0.3, 0.6, 0.9) and Sp = (0.6, 0.9). The proportions of verification $P(V = 1)$ were 0.54, 0.47, 0.59, 0.52, 0.64 and 0.57 for the (Sn, Sp) pairs (0.3, 0.6), (0.3, 0.9), (0.6, 0.6), (0.6, 0.9), (0.9, 0.6) and (0.9, 0.9) respectively. The best values (i.e. the smallest bias and SE values) achieved by IPB, SIPW, and SIPW-B are marked with an asterisk.

**Table 1. Comparison between IPB, SIPW, SIPW-B and existing PVB correction methods for $p = 0.4$ with $n = 200$ and $1000$ under six combinations of Sn and Sp.**

| Methods | Mean | Bias | SE | Mean | Bias | SE | Mean | Bias | SE | Mean | Bias | SE |
|---|---|---|---|---|---|---|---|---|---|---|---|---|
| | $n = 200$ | | | | | | $n = 1000$ | | | | | |
| | Sn = 0.3 | | | Sp = 0.6 | | | Sn = 0.3 | | | Sp = 0.6 | | |
| FDA | 0.302 | 0.002 | 0.050 | 0.603 | 0.003 | 0.044 | 0.302 | 0.002 | 0.023 | 0.600 | 0.000 | 0.019 |
| CCA | 0.465 | 0.165 | 0.076 | 0.430 | −0.170 | 0.060 | 0.465 | 0.165 | 0.036 | 0.429 | −0.171 | 0.027 |
| BG | 0.303 | 0.003 | 0.058 | 0.602 | 0.002 | 0.051 | 0.303 | 0.003 | 0.028 | 0.601 | 0.001 | 0.023 |
| IPWE | 0.303 | 0.003 | 0.058 | 0.602 | 0.002 | 0.051 | 0.303 | 0.003 | 0.028 | 0.601 | 0.001 | 0.023 |
| MI | 0.305 | 0.005 | 0.060 | 0.600 | 0.000 | 0.053 | 0.303 | 0.003 | 0.029 | 0.600 | 0.000 | 0.023 |
| IPB | 0.300 | *0.000 | 0.085 | 0.596 | *−0.004 | 0.080 | 0.302 | *0.002 | 0.042 | 0.600 | *0.000 | 0.035 |
| SIPW | 0.302 | 0.002 | 0.079 | 0.604 | *0.004 | 0.070 | 0.302 | *0.002 | 0.036 | 0.601 | 0.001 | *0.031 |
| SIPW-B | 0.302 | 0.002 | *0.078 | 0.605 | 0.005 | *0.067 | 0.304 | 0.004 | *0.035 | 0.602 | 0.002 | 0.032 |
| | Sn = 0.3 | | | Sp = 0.9 | | | Sn = 0.3 | | | Sp = 0.9 | | |
| FDA | 0.302 | 0.002 | 0.050 | 0.902 | 0.002 | 0.027 | 0.302 | 0.002 | 0.023 | 0.900 | 0.000 | 0.013 |
| CCA | 0.466 | 0.166 | 0.077 | 0.821 | −0.079 | 0.053 | 0.464 | 0.164 | 0.034 | 0.818 | −0.082 | 0.024 |
| BG | 0.305 | 0.005 | 0.060 | 0.902 | 0.002 | 0.030 | 0.302 | 0.002 | 0.026 | 0.900 | 0.000 | 0.014 |
| IPWE | 0.305 | 0.005 | 0.060 | 0.902 | 0.002 | 0.030 | 0.302 | 0.002 | 0.026 | 0.900 | 0.000 | 0.014 |
| MI | 0.306 | 0.006 | 0.061 | 0.901 | 0.001 | 0.030 | 0.302 | 0.002 | 0.027 | 0.900 | 0.000 | 0.014 |
| IPB | 0.306 | *0.006 | 0.095 | 0.903 | 0.003 | 0.050 | 0.301 | *0.001 | 0.043 | 0.900 | *0.000 | 0.022 |
| SIPW | 0.307 | 0.007 | 0.081 | 0.902 | *0.002 | *0.041 | 0.302 | 0.002 | 0.035 | 0.901 | 0.001 | *0.018 |
| SIPW-B | 0.308 | 0.008 | *0.076 | 0.905 | 0.005 | 0.042 | 0.301 | *0.001 | *0.034 | 0.900 | *0.000 | 0.019 |
| | Sn = 0.6 | | | Sp = 0.6 | | | Sn = 0.6 | | | Sp = 0.6 | | |
| FDA | 0.603 | 0.003 | 0.055 | 0.602 | 0.002 | 0.044 | 0.602 | 0.002 | 0.023 | 0.600 | 0.000 | 0.019 |
| CCA | 0.754 | 0.154 | 0.060 | 0.430 | −0.170 | 0.060 | 0.751 | 0.151 | 0.026 | 0.428 | −0.172 | 0.026 |
| BG | 0.607 | 0.007 | 0.072 | 0.602 | 0.002 | 0.050 | 0.602 | 0.002 | 0.031 | 0.600 | 0.000 | 0.022 |
| IPWE | 0.607 | 0.007 | 0.072 | 0.602 | 0.002 | 0.050 | 0.602 | 0.002 | 0.031 | 0.600 | 0.000 | 0.022 |
| MI | 0.605 | 0.005 | 0.075 | 0.599 | −0.001 | 0.052 | 0.601 | 0.001 | 0.032 | 0.599 | −0.001 | 0.022 |
| IPB | 0.609 | 0.009 | 0.105 | 0.602 | 0.002 | 0.078 | 0.599 | *−0.001 | 0.044 | 0.601 | 0.001 | 0.034 |
| SIPW | 0.608 | 0.008 | 0.090 | 0.603 | 0.003 | *0.065 | 0.604 | 0.004 | *0.037 | 0.600 | *0.000 | *0.029 |
| SIPW-B | 0.606 | *0.006 | *0.084 | 0.600 | *0.000 | 0.070 | 0.602 | 0.002 | 0.038 | 0.600 | *0.000 | 0.031 |
| | Sn = 0.6 | | | Sp = 0.9 | | | Sn = 0.6 | | | Sp = 0.9 | | |
| FDA | 0.603 | 0.003 | 0.055 | 0.902 | 0.002 | 0.027 | 0.602 | 0.002 | 0.023 | 0.900 | 0.000 | 0.012 |
| CCA | 0.754 | 0.154 | 0.061 | 0.822 | −0.078 | 0.054 | 0.752 | 0.152 | 0.026 | 0.818 | −0.082 | 0.025 |
| BG | 0.608 | 0.008 | 0.075 | 0.903 | 0.003 | 0.030 | 0.602 | 0.002 | 0.031 | 0.900 | 0.000 | 0.014 |
| IPWE | 0.608 | 0.008 | 0.075 | 0.903 | 0.003 | 0.030 | 0.602 | 0.002 | 0.031 | 0.900 | 0.000 | 0.014 |
| MI | 0.605 | 0.005 | 0.076 | 0.901 | 0.001 | 0.031 | 0.602 | 0.002 | 0.032 | 0.900 | 0.000 | 0.014 |
| IPB | 0.605 | *0.005 | 0.118 | 0.903 | *0.003 | 0.049 | 0.599 | *−0.001 | 0.048 | 0.899 | −0.001 | 0.024 |
| SIPW | 0.610 | 0.010 | 0.094 | 0.903 | *0.003 | *0.040 | 0.604 | 0.004 | 0.040 | 0.901 | 0.001 | *0.019 |
| SIPW-B | 0.608 | 0.008 | *0.086 | 0.905 | 0.005 | 0.042 | 0.602 | 0.002 | *0.039 | 0.900 | *0.000 | 0.020 |
| | Sn = 0.9 | | | Sp = 0.6 | | | Sn = 0.9 | | | Sp = 0.6 | | |
| FDA | 0.899 | −0.001 | 0.033 | 0.601 | 0.001 | 0.044 | 0.901 | 0.001 | 0.015 | 0.601 | 0.001 | 0.020 |
| CCA | 0.945 | 0.045 | 0.027 | 0.427 | −0.173 | 0.057 | 0.948 | 0.048 | 0.012 | 0.429 | −0.171 | 0.027 |
| BG | 0.896 | −0.004 | 0.046 | 0.600 | 0.000 | 0.046 | 0.901 | 0.001 | 0.022 | 0.601 | 0.001 | 0.021 |
| IPWE | 0.896 | −0.004 | 0.046 | 0.600 | 0.000 | 0.046 | 0.901 | 0.001 | 0.022 | 0.601 | 0.001 | 0.021 |
| MI | 0.889 | −0.011 | 0.046 | 0.598 | −0.002 | 0.047 | 0.899 | −0.001 | 0.023 | 0.600 | 0.000 | 0.021 |
| IPB | 0.894 | −0.006 | 0.064 | 0.595 | −0.005 | 0.072 | 0.901 | *0.001 | 0.028 | 0.600 | *0.000 | 0.033 |
| SIPW | 0.896 | −0.004 | 0.058 | 0.597 | −0.003 | *0.064 | 0.901 | *0.001 | 0.027 | 0.600 | *0.000 | *0.029 |
| SIPW-B | 0.897 | *−0.003 | *0.055 | 0.599 | *−0.001 | 0.069 | 0.901 | *0.001 | *0.025 | 0.603 | 0.003 | 0.031 |
| | Sn = 0.9 | | | Sp = 0.9 | | | Sn = 0.9 | | | Sp = 0.9 | | |
| FDA | 0.899 | −0.001 | 0.033 | 0.900 | 0.000 | 0.028 | 0.901 | 0.001 | 0.015 | 0.901 | 0.001 | 0.012 |
| CCA | 0.946 | 0.046 | 0.028 | 0.817 | −0.083 | 0.054 | 0.948 | 0.048 | 0.013 | 0.819 | −0.081 | 0.024 |
| BG | 0.899 | −0.001 | 0.048 | 0.900 | 0.000 | 0.030 | 0.901 | 0.001 | 0.022 | 0.901 | 0.001 | 0.013 |
| IPWE | 0.899 | −0.001 | 0.048 | 0.900 | 0.000 | 0.030 | 0.901 | 0.001 | 0.022 | 0.901 | 0.001 | 0.013 |
| MI | 0.889 | −0.011 | 0.050 | 0.898 | −0.002 | 0.031 | 0.899 | −0.001 | 0.023 | 0.901 | 0.001 | 0.013 |
| IPB | 0.899 | −0.001 | 0.066 | 0.901 | 0.001 | 0.045 | 0.901 | *0.001 | 0.031 | 0.901 | *0.001 | 0.020 |
| SIPW | 0.900 | *0.000 | 0.060 | 0.900 | *0.000 | *0.041 | 0.902 | 0.002 | *0.026 | 0.901 | *0.001 | *0.019 |
| SIPW-B | 0.898 | −0.002 | *0.057 | 0.898 | −0.002 | 0.044 | 0.902 | 0.002 | *0.026 | 0.901 | *0.001 | *0.019 |

Abbreviations: BG, Begg and Greenes' method; CCA, complete case analysis; FDA, Full data analysis; IPB, inverse probability bootstrap; IPWE, inverse probability weighting estimator; MI, multiple imputation; $n$, sample size; $p$, disease prevalence; SE, standard error; SIPW, scaled inverse probability weighted resampling; SIPW-B, scaled inverse probability weighted balanced resampling; Sn, sensitivity; Sp, specificity.

Next, The simulation results for FDA, CCA and the PVB correction methods for $p = 0.1$ are displayed in Table 2 for sample sizes $n = 200$ and 1000. The results are organized by parameter combinations of Sn = (0.3, 0.6, 0.9) and Sp = (0.6, 0.9). The proportions of verification $P(V = 1)$ were 0.56, 0.45, 0.57, 0.46, 0.58 and 0.47 for the (Sn, Sp) pairs (0.3, 0.6), (0.3, 0.9), (0.6, 0.6), (0.6, 0.9), (0.9, 0.6) and (0.9, 0.9) respectively. Again, the smallest bias and SE values achieved by IPB, SIPW, and SIPW-B are marked with an asterisk.

As observed in Tables 1 and 2, both SIPW and SIPW-B generally performed better than IPB for Sn and Sp estimation, as indicated by smaller bias and SE values. Bias results were mixed, with only marginal differences among IPB, SIPW and SIPW-B. However, SIPW and SIPW-B most often showed slightly lower bias than IPB. SIPW-B showed smaller SE for Sn estimation compared to SIPW because it enlarges the size of the case group ($n_{D=1}$). In contrast, SIPW showed smaller SE than SIPW-B for Sp estimation because it maintains the original size of the control group ($n_{D=0}$), which is larger than the case group when disease prevalence is $p = 0.4$ (Table 1). This effect was more pronounced at a lower disease prevalence of $p = 0.1$ (Table 2). When compared to existing methods (BG, IPWE, and MI), both new methods closely matched their performance in terms of bias and SE. Another observation was that, across all PVB correction methods, both bias and SE decreased as disease prevalence and sample size increased. Counterintuitively, when prevalence was low and sample size was small, CCA showed less bias than FDA and other PVB correction methods at a very high Sn value of 0.9.

## Clinical data analysis

The results comparing CCA (bias uncorrected) and the PVB correction methods using the clinical data sets are displayed in Table 3. Across these data sets, all PVB correction methods showed nearly identical point estimates for Sn and Sp, except for MI, which showed a slightly different Sn estimate for the diaphanography data set. For this data set, SIPW and SIPW-B showed 95% CIs consistent with existing methods. Specifically, for Sn, three PS-based methods (IPWE, SIPW, and SIPW-B) showed almost identical 95% CIs, while BG and MI were similar to each other. In contrast, IPB exhibited notably wider 95% CIs for both Sn and Sp, differing from the estimates obtained by SIPW, SIPW-B and the existing methods.

## Discussion

To address the limitations of IPB, this study introduces two new methods based on IPB: SIPW and SIPW-B. The first limitation of IPB is that, although it showed small bias in estimating Sn and Sp, it had relatively larger SE than existing methods (BG, IPWE and MI). The second limitation of IPB is that IPB only corrects the distribution of the verified portion of the PVB data. This study demonstrated that the proposed methods successfully overcome these limitations, as discussed in detail below.

From the simulated data analysis, both SIPW and SIPW-B showed smaller bias and SE than IPB for Sn and Sp estimation, although the differences in bias were less pronounced. SIPW-B showed the lowest SE among these three methods for Sn estimation, while SIPW showed the lowest SE for Sp estimation. The new methods also matched the existing methods by showing low bias and SE for both Sn and Sp estimation. In contrast, as noted in [21], a major drawback of IPB is its large SE compared to other existing methods. SIPW and SIPW-B overcame this issue and demonstrated SE comparable to the existing methods. Although both methods performed better than IPB, SIPW performed better than SIPW-B for Sp estimation, while SIPW-B performed better than SIPW for Sn estimation. SIPW-B was designed to mimic the case-control study design. This approach is similar to common machine learning

**Table 2. Comparison between IPB, SIPW, SIPW-B and existing PVB correction methods for $p = 0.1$ with $n = 200$ and $1000$ under six combinations of Sn and Sp.**

| Methods | Mean | Bias | SE | Mean | Bias | SE | Mean | Bias | SE | Mean | Bias | SE |
|---|---|---|---|---|---|---|---|---|---|---|---|---|
| | $n = 200$ | | | | | | $n = 1000$ | | | | | |
| | Sn = 0.3 | | | Sp = 0.6 | | | Sn = 0.3 | | | Sp = 0.6 | | |
| FDA | 0.301 | 0.001 | 0.107 | 0.601 | 0.001 | 0.037 | 0.300 | 0.000 | 0.043 | 0.600 | 0.000 | 0.016 |
| CCA | 0.459 | 0.159 | 0.159 | 0.428 | −0.172 | 0.050 | 0.462 | 0.162 | 0.066 | 0.429 | −0.171 | 0.022 |
| BG | 0.310 | 0.010 | 0.141 | 0.600 | 0.000 | 0.039 | 0.302 | 0.002 | 0.055 | 0.600 | 0.000 | 0.017 |
| IPWE | 0.310 | 0.010 | 0.141 | 0.600 | 0.000 | 0.039 | 0.302 | 0.002 | 0.055 | 0.600 | 0.000 | 0.017 |
| MI | 0.303 | 0.003 | 0.133 | 0.598 | −0.002 | 0.039 | 0.303 | 0.003 | 0.055 | 0.600 | 0.000 | 0.017 |
| IPB | 0.309 | 0.009 | 0.209 | 0.601 | *0.001 | 0.063 | 0.303 | 0.003 | 0.080 | 0.600 | *0.000 | 0.027 |
| SIPW | 0.306 | *0.006 | 0.167 | 0.599 | *−0.001 | *0.052 | 0.308 | 0.008 | 0.071 | 0.599 | −0.001 | *0.024 |
| SIPW-B | 0.307 | 0.007 | *0.148 | 0.601 | *0.001 | 0.060 | 0.300 | *0.000 | *0.058 | 0.599 | −0.001 | 0.028 |
| | Sn = 0.3 | | | Sp = 0.9 | | | Sn = 0.3 | | | Sp = 0.9 | | |
| FDA | 0.298 | −0.002 | 0.106 | 0.901 | 0.001 | 0.021 | 0.300 | 0.000 | 0.043 | 0.900 | 0.000 | 0.010 |
| CCA | 0.464 | 0.164 | 0.160 | 0.819 | −0.081 | 0.042 | 0.461 | 0.161 | 0.068 | 0.818 | −0.082 | 0.018 |
| BG | 0.314 | 0.014 | 0.136 | 0.901 | 0.001 | 0.022 | 0.302 | 0.002 | 0.056 | 0.900 | 0.000 | 0.010 |
| IPWE | 0.314 | 0.014 | 0.136 | 0.901 | 0.001 | 0.022 | 0.302 | 0.002 | 0.056 | 0.900 | 0.000 | 0.010 |
| MI | 0.302 | 0.002 | 0.129 | 0.901 | 0.001 | 0.023 | 0.299 | −0.001 | 0.059 | 0.900 | 0.000 | 0.010 |
| IPB | 0.323 | 0.023 | 0.227 | 0.903 | 0.003 | 0.039 | 0.305 | 0.005 | 0.092 | 0.900 | *0.000 | 0.019 |
| SIPW | 0.310 | *0.010 | 0.178 | 0.901 | *0.001 | *0.033 | 0.304 | 0.004 | 0.074 | 0.900 | *0.000 | *0.014 |
| SIPW-B | 0.318 | 0.018 | *0.143 | 0.904 | 0.004 | 0.037 | 0.302 | *0.002 | *0.061 | 0.900 | *0.000 | 0.017 |
| | Sn = 0.6 | | | Sp = 0.6 | | | Sn = 0.6 | | | Sp = 0.6 | | |
| FDA | 0.596 | −0.004 | 0.112 | 0.601 | 0.001 | 0.037 | 0.600 | 0.000 | 0.048 | 0.600 | 0.000 | 0.017 |
| CCA | 0.743 | 0.143 | 0.117 | 0.429 | −0.171 | 0.049 | 0.754 | 0.154 | 0.053 | 0.429 | −0.171 | 0.022 |
| BG | 0.603 | 0.003 | 0.144 | 0.601 | 0.001 | 0.038 | 0.607 | 0.007 | 0.067 | 0.601 | 0.001 | 0.017 |
| IPWE | 0.603 | 0.003 | 0.144 | 0.601 | 0.001 | 0.038 | 0.607 | 0.007 | 0.067 | 0.601 | 0.001 | 0.017 |
| MI | 0.579 | −0.021 | 0.136 | 0.598 | −0.002 | 0.039 | 0.601 | 0.001 | 0.067 | 0.600 | 0.000 | 0.017 |
| IPB | 0.595 | −0.005 | 0.202 | 0.606 | 0.006 | 0.062 | 0.605 | 0.005 | 0.093 | 0.600 | *0.000 | 0.027 |
| SIPW | 0.613 | 0.013 | 0.184 | 0.603 | 0.003 | *0.055 | 0.604 | *0.004 | 0.084 | 0.600 | *0.000 | *0.025 |
| SIPW-B | 0.600 | *0.000 | *0.157 | 0.601 | *0.001 | 0.062 | 0.607 | 0.007 | *0.071 | 0.601 | 0.001 | 0.027 |
| | Sn = 0.6 | | | Sp = 0.9 | | | Sn = 0.6 | | | Sp = 0.9 | | |
| FDA | 0.600 | 0.000 | 0.115 | 0.900 | 0.000 | 0.022 | 0.600 | 0.000 | 0.048 | 0.900 | 0.000 | 0.010 |
| CCA | 0.738 | 0.138 | 0.119 | 0.818 | −0.082 | 0.043 | 0.749 | 0.149 | 0.052 | 0.819 | −0.081 | 0.019 |
| BG | 0.598 | −0.002 | 0.143 | 0.900 | 0.000 | 0.023 | 0.601 | 0.001 | 0.065 | 0.900 | 0.000 | 0.010 |
| IPWE | 0.598 | −0.002 | 0.143 | 0.900 | 0.000 | 0.023 | 0.601 | 0.001 | 0.065 | 0.900 | 0.000 | 0.010 |
| MI | 0.568 | −0.032 | 0.134 | 0.900 | 0.000 | 0.023 | 0.593 | −0.007 | 0.065 | 0.900 | 0.000 | 0.010 |
| IPB | 0.599 | −0.001 | 0.214 | 0.898 | −0.002 | 0.042 | 0.604 | 0.004 | 0.099 | 0.901 | 0.001 | 0.018 |
| SIPW | 0.600 | *0.000 | 0.178 | 0.899 | *−0.001 | *0.033 | 0.602 | 0.002 | 0.080 | 0.901 | 0.001 | *0.014 |
| SIPW-B | 0.598 | −0.002 | *0.152 | 0.902 | 0.002 | 0.038 | 0.600 | *0.000 | *0.067 | 0.900 | *0.000 | 0.017 |
| | Sn = 0.9 | | | Sp = 0.6 | | | Sn = 0.9 | | | Sp = 0.6 | | |
| FDA | 0.875 | −0.025 | 0.062 | 0.602 | 0.002 | 0.035 | 0.899 | −0.001 | 0.028 | 0.600 | 0.000 | 0.016 |
| CCA | 0.910 | 0.010 | 0.042 | 0.430 | −0.170 | 0.048 | 0.947 | 0.047 | 0.025 | 0.428 | −0.172 | 0.021 |
| BG | 0.837 | −0.063 | 0.068 | 0.599 | −0.001 | 0.037 | 0.900 | 0.000 | 0.044 | 0.600 | 0.000 | 0.017 |
| IPWE | 0.837 | −0.063 | 0.068 | 0.599 | −0.001 | 0.037 | 0.900 | 0.000 | 0.044 | 0.600 | 0.000 | 0.017 |
| MI | 0.800 | −0.100 | 0.080 | 0.597 | −0.003 | 0.037 | 0.889 | −0.011 | 0.046 | 0.600 | 0.000 | 0.017 |
| IPB | 0.832 | −0.068 | 0.133 | 0.605 | 0.005 | 0.059 | 0.897 | −0.003 | 0.060 | 0.601 | 0.001 | 0.027 |
| SIPW | 0.837 | *−0.063 | 0.108 | 0.600 | *0.000 | *0.050 | 0.901 | 0.001 | 0.052 | 0.600 | *0.000 | *0.023 |
| SIPW-B | 0.837 | *−0.063 | *0.077 | 0.600 | *0.000 | 0.061 | 0.900 | *0.000 | *0.045 | 0.601 | 0.001 | 0.028 |
| | Sn = 0.9 | | | Sp = 0.9 | | | Sn = 0.9 | | | Sp = 0.9 | | |
| FDA | 0.874 | −0.026 | 0.062 | 0.901 | 0.001 | 0.023 | 0.900 | 0.000 | 0.029 | 0.900 | 0.000 | 0.010 |
| CCA | 0.905 | 0.005 | 0.051 | 0.819 | −0.081 | 0.045 | 0.948 | 0.048 | 0.025 | 0.819 | −0.081 | 0.020 |
| BG | 0.830 | −0.070 | 0.079 | 0.900 | 0.000 | 0.025 | 0.901 | 0.001 | 0.044 | 0.900 | 0.000 | 0.011 |
| IPWE | 0.830 | −0.070 | 0.079 | 0.900 | 0.000 | 0.025 | 0.901 | 0.001 | 0.044 | 0.900 | 0.000 | 0.011 |
| MI | 0.777 | −0.123 | 0.087 | 0.899 | −0.001 | 0.025 | 0.887 | −0.013 | 0.045 | 0.900 | 0.000 | 0.011 |
| IPB | 0.829 | −0.071 | 0.158 | 0.900 | *0.000 | 0.044 | 0.898 | −0.002 | 0.061 | 0.900 | *0.000 | 0.018 |
| SIPW | 0.831 | −0.069 | 0.112 | 0.900 | *0.000 | *0.034 | 0.901 | *0.001 | 0.055 | 0.900 | *0.000 | *0.015 |
| SIPW-B | 0.832 | *−0.068 | *0.089 | 0.900 | *0.000 | 0.037 | 0.901 | *0.001 | *0.046 | 0.900 | *0.000 | 0.016 |

Abbreviations: BG, Begg and Greenes' method; CCA, complete case analysis; FDA, Full data analysis; IPB, inverse probability bootstrap; IPWE, inverse probability weighting estimator; MI, multiple imputation; $n$, sample size; $p$, disease prevalence; SE, standard error; SIPW, scaled inverse probability weighted resampling; SIPW-B, scaled inverse probability weighted balanced resampling; Sn, sensitivity; Sp, specificity.

**Table 3. Sn and Sp estimates of IPB, SIPW, SIPW-B and other methods with the respective 95% CIs using clinical data sets.**

| Methods | Scintigraphy data set | | Diaphanography data set | |
|---|---|---|---|---|
| | Sn (95% CI) | Sp (95% CI) | Sn (95% CI) | Sp (95% CI) |
| CCA | 0.895 (0.858, 0.933) | 0.628 (0.526, 0.730) | 0.788 (0.648, 0.927) | 0.800 (0.694, 0.906) |
| BG | 0.836 (0.788, 0.884) | 0.738 (0.662, 0.815) | 0.292 (0.134, 0.449) | 0.973 (0.958, 0.988) |
| IPWE | 0.836 (0.785, 0.885) | 0.738 (0.656, 0.812) | 0.292 (0.177, 0.548) | 0.973 (0.957, 0.987) |
| MI | 0.834 (0.782, 0.885) | 0.738 (0.661, 0.815) | 0.279 (0.124, 0.435) | 0.972 (0.957, 0.987) |
| IPB | 0.838 (0.793, 0.881) | 0.738 (0.650, 0.824) | 0.290 (0.077, 0.529) | 0.973 (0.931, 1.000) |
| SIPW | 0.837 (0.785, 0.886) | 0.739 (0.655, 0.811) | 0.292 (0.176, 0.548) | 0.973 (0.957, 0.987) |
| SIPW-B | 0.837 (0.785, 0.885) | 0.739 (0.655, 0.812) | 0.291 (0.176, 0.548) | 0.973 (0.957, 0.987) |

Abbreviations: BG, Begg and Greenes' method; CCA, complete case analysis; CI, confidence interval; IPB, inverse probability bootstrap; IPWE, inverse probability weighting estimator; MI, multiple imputation; SIPW, scaled inverse probability weighted resampling; SIPW-B, scaled inverse probability weighted balanced resampling; Sn, sensitivity; Sp, specificity.

techniques that handle class imbalance by rebalancing the class distribution through over- or under-sampling [33,49–51]. SIPW-B balances the effect of subgroup sample sizes by resizing them to a predefined ratio of $n_{D=0}:n_{D=1}$, while keeping the original sample size $n = n_{D=0} + n_{D=1}$. The trade-off is that it lowers the precision of the larger subgroup, although the precision for the smaller subgroup increases. For example, for a sample size $n = 1000$ with $p = 0.2$, the subgroup sizes are $n_{D=1} = n \times p = 200$ and $n_{D=0} = n \times (1-p) = 800$. If the desired control:case ratio is 1:1, before balancing, Sn is calculated from $n_{D=1} = 200$ and Sp from $n_{D=0} = 800$. After balancing, Sn is calculated from $n_{D=1} = 500$ and Sp from $n_{D=0} = 500$. Notably, $n_{D=0}$ drops from 800 to 500 for Sp calculation, reducing both numerator and denominator counts, which lowers Sp precision. This pattern was also observed in [50], where random under-sampling degraded the performance of machine learning methods, mainly due to information loss from discarding observations [51]. On the other hand, $n_{D=1}$ increases from 200 to 500 for Sn calculation, increasing both numerator and denominator counts and improving Sn precision. In contrast, SIPW keeps $n_{D=0}$ unchanged for Sp estimation. In addition, after applying SIPW-B, statistics that rely on the true distribution of $P(D = 1)$ (such as PPV = $P[D = 1|T = 1]$ or NPV = $P[D = 0|T = 0]$) are no longer valid [8]. Although PPV and NPV can be calculated indirectly from Sn and Sp estimates of SIPW-B by utilizing Bayes' theorem [3,11], further research is needed to study this implementation. Therefore, SIPW-B is recommended when only Sn and Sp are the main estimates. On the other hand, SIPW is recommended when full data restoration is needed for further analysis involving metrics that depend on prevalence, such as PPV, NPV and accuracy [3,52].

At low prevalence settings, MI generally exhibited the largest bias in Sn estimation compared to other correction methods. The proposed methods showed lower bias than MI and were comparable to BG and IPWE in terms of bias, while showing smaller SE than IPB. As explained in a previous study [21], IPB showed larger SE because it only resamples verified observations. When prevalence $p$ (i.e., $P(D = 1)$) is low, this limits the available observations for estimating Sn to $n_{D=1} = p \times P(V = 1) \times n$. The proposed methods do not have this limitation because they restore the complete sample by resampling both verified and unverified observations, making the available observations for estimating Sn equal to $n_{D=1} = p \times n$, which is no longer dependent on $V = 1$. For practical applications, when disease prevalence is low, SIPW-B is preferable for estimating Sn, whereas SIPW is better suited for Sp estimation. The higher bias shown by MI at low prevalence is unexpected, given that MI also restores the complete sample. Although this may be related to MI's reliance on accurate estimation of

$P(D = d|T = t)$, low prevalence did not affect the BG method, which also depends on this probability. This suggests the need for a further study to investigate the performance of MI-based PVB correction methods, particularly under different experimental conditions and imputation strategies [45,53]. Although Day et al. [15] examined an extension of the BG method and two MI-based methods, this issue was not observed because their proposed methods were not tested on simulated data sets.

In the clinical data sets, the results varied by data set. For the scintigraphy data set, all correction methods generally showed consistent results with each other. In contrast, for the diaphanography data set, the point estimate of Sn for MI differed from other methods, which could be explained by previously discussed simulation findings. SIPW and SIPW-B showed 95% CIs comparable to existing methods, whereas IPB exhibited notably wider 95% CIs for both Sn and Sp, diverging from those estimated by other methods. Given the small observed sample size for this data set (only 88 verified out of 900 patients), IPB appears to perform poorly for interval estimation when data are limited [21]. This could also be attributed to its large SE, as demonstrated in the simulated data sets. Since smaller sample sizes are generally associated with larger SE [21,44], this finding is expected and was previously noted in [21]. SIPW and SIPW-B, which incorporate full data restoration, performed better than IPB in this condition. The observation that SIPW and SIPW-B performed comparably to existing methods is consistent with findings by Day et al. [15], who reported good performance of MI-based approaches, also employing full data restoration, on a clinical dataset.

The correction methods examined in this study differ in the probabilities they estimate relative to the test result. PS-based methods (IPWE, IPB, SIPW and SIPW-B) rely on PS, the probability of verification given the test result, $P(V = 1|T = t)$. This probability is directly related to the verification problem, particularly the PVB problem [21]. PS is used as the weight to adjust for the verification bias, resulting in corrected Sn and Sp estimates [54]. Reweighting is also a recommended strategy in mitigating bias in machine learning [55]. While BG-based and MI methods rely on accurate estimation of the probability of disease status given the test result, $P(D = d|T = t)$ [15], PS-based methods instead rely on accurate estimation of $P(V = 1|T = t)$ to perform the correction [11,55]. This makes PS-based methods particularly advantageous in diagnostic accuracy studies employing a case-control design, as $P(D = d|T = t)$ may be inaccurately estimated in this situation [8]. Wang et al. [54] recently showed that their methods based on IPWE, a PS-based method, demonstrated good performance. In the diaphanography data set, using either $P(V = 1|T = t)$ or $P(D = d|T = t)$) to correct for bias led to different Sn estimates. Three PS-based methods (IPWE, SIPW, and SIPW-B), which rely on $P(V = 1|T = t)$, showed nearly identical 95% CIs. In contrast, BG and MI, which rely on $P(D = d|T = t)$, showed similar 95% CIs to each other. This finding suggests that further investigation is needed to evaluate the conditions under which these approaches produce different results.

The correction methods in this study also differ by the stage where bias correction occurs, either at the data level or the algorithm level [49,51,56,57]. The data-level methods in this study are MI, IPB, SIPW, and SIPW-B, while the algorithm-level methods are BG and IPWE. Data-level methods modify the data itself, which allows the use of any existing analytical methods or combinations of algorithms [20,57]. In contrast, algorithm-level methods require adapting or creating specific algorithms for specific use cases, but they are more computationally efficient and easier to apply than data-level methods [20,57]. MI, IPB, SIPW, and SIPW-B apply data-level bias correction [56] by restoring the sample distribution of data affected by PVB, which allows further analysis using any complete-data methods [20,45]. These four

methods only differ in how they restore the data, while Sn and Sp can be calculated from standard formulas for these accuracy measures [11,24]. This was shown in [58] for the kappa coefficient in diagnostic accuracy studies and in [15] for PVB correction in multiple-test situations. If an algorithm-level approach had been used in [58] and [15], it would have required specific algorithms. For this reason, Day et al. [15] extended the existing BG method for their study, while MI was easily applied using an existing MI method [24]. Among the four data-level methods, IPB restores only the portion of data containing verified observations ($n_1$) [20, 21], whereas SIPW, SIPW-B, and MI [15,24] restore the full data containing both verified and unverified observations ($n$). As seen in the simulated and clinical datasets, this reduced the precision of IPB, shown by larger SE and wider confidence intervals. Compared to MI, SIPW and SIPW-B are easier to apply because they only require estimating PS values, followed by performing a weighted resampling procedure. In contrast, MI for PVB correction requires selecting suitable imputation methods, as its performance in bias correction depends on the chosen method [15,45,53].

While this study has demonstrated the strengths of SIPW and SIPW-B, these methods present two notable limitations. First, they are computationally intensive methods because they rely on repeated resampling. For example, with $b = 1000$ samples, they require substantially more computational resources to perform resampling 1000 times, whereas algorithm-level methods typically require only a single iteration. Second, unlike IPB, CIs cannot be derived directly from these resamples because they no longer form valid bootstrap samples. For interval estimation, an additional computationally demanding bootstrapping procedure must be performed on top of the algorithms. Specifically, if 1000 bootstrap replicates ($R = 1000$) are required for interval estimation, the computational burden increases significantly. For instance, if 1000 bootstrap replicates ($R = 1000$) are needed for interval estimation, the time taken to complete a full SIPW procedure (e.g., 10 seconds with $b = 1000$ SIPW samples) will be multiplied by 1000.

## Conclusion

This paper proposes the SIPW and SIPW-B methods to address the limitations of the IPB method in the context of PVB correction under the MAR assumption for binary diagnostic tests. The results show that both SIPW and SIPW-B outperformed IPB and were consistent with existing PVB correction methods. Specifically, SIPW excelled in estimating Sp, while SIPW-B performed best in Sn estimation. The proposed methods also demonstrated good performance when disease prevalence was low. In addition, they improve upon IPB by allowing full data restoration, enabling subsequent analysis using any complete-data analytical methods. Although the new methods currently require more computational resources, this is expected to become less of an issue with advancements in computational power.

## Acknowledgments

We thank our colleagues at the School of Computer Sciences and the School of Medical Sciences, Universiti Sains Malaysia for their comments on the early findings of this study and this article's draft.

## Author contributions

**Conceptualization:** Wan Nor Arifin, Umi Kalsom Yusof.

**Data curation:** Wan Nor Arifin.

**Formal analysis:** Wan Nor Arifin.

**Funding acquisition:** Wan Nor Arifin, Umi Kalsom Yusof.

**Investigation:** Wan Nor Arifin.

**Methodology:** Wan Nor Arifin, Umi Kalsom Yusof.

**Project administration:** Wan Nor Arifin.

**Resources:** Wan Nor Arifin, Umi Kalsom Yusof.

**Software:** Wan Nor Arifin.

**Supervision:** Umi Kalsom Yusof.

**Validation:** Wan Nor Arifin, Umi Kalsom Yusof.

**Writing – original draft:** Wan Nor Arifin.

**Writing – review & editing:** Wan Nor Arifin, Umi Kalsom Yusof.

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
