## [Decision Letter · Decision Letter 0]

11 Jun 2025

PONE-D-25-12005

Partial Verification Bias Correction Using Scaled Inverse Probability Resampling for Binary Diagnostic Tests

PLOS ONE

Dear Dr. Arifin,

Thank you for submitting your manuscript to PLOS ONE. After careful consideration, we feel that it has merit but does not fully meet PLOS ONE’s publication criteria as it currently stands. Therefore, we invite you to submit a revised version of the manuscript that addresses the points raised during the review process.

We look forward to receiving your revised manuscript.

Kind regards,

Hayrunnisa Nadaroglu

Academic Editor

PLOS ONE

Journal Requirements:

3. Please note that your Data Availability Statement is currently missing the repository name and/or the DOI/accession number of each dataset OR a direct link to access each database. If your manuscript is accepted for publication, you will be asked to provide these details on a very short timeline. We therefore suggest that you provide this information now, though we will not hold up the peer review process if you are unable.

Reviewers' comments:

Reviewer's Responses to Questions

**Comments to the Author**

1. Is the manuscript technically sound, and do the data support the conclusions?

Reviewer #1: Yes

Reviewer #2: Yes

Reviewer #3: Yes

2. Has the statistical analysis been performed appropriately and rigorously? 

Reviewer #1: Yes

Reviewer #2: Yes

Reviewer #3: Yes

3. Have the authors made all data underlying the findings in their manuscript fully available?

Reviewer #1: Yes

Reviewer #2: Yes

Reviewer #3: Yes

4. Is the manuscript presented in an intelligible fashion and written in standard English?

Reviewer #1: Yes

Reviewer #2: Yes

Reviewer #3: Yes

5. Review Comments to the Author

Reviewer #1: “Partial Verification Bias Correction Using Scaled Inverse Probability Resampling for Binary Diagnostic Tests”

Minor corrections to the article titled are listed below;

1-The summary should be expanded a bit.

2- The last sentence in the summary is not understandable, it should be rewritten

3-The spelling of the “universiti” in the address section should be corrected.

4-The sentence starting with "in the study" on line 34 and the rest should be written in a new paragraph.

5-The word "ın words" in line 62 should be corrected to "In the other words".

6-The sentence starting with "In addition" in line 171 is not understood.

7-"The side effect" in line 207 this term is used in the medical field. Please write the sentence with another word.

8- A word expressing possibility should be used instead of "should" in line 328.

9-The spelling of the word “universiti” in line 339 should be corrected.

10- Article spelling and grammar rules should be reviewed.

Reviewer #2: The study aimed to reveal the importance of performing diagnostic accuracy tests to evaluate new tests before clinical tests.

These tests, where accuracy measurements such as sensitivity (Sn) and specificity (Sp) are frequently used, are compared with standard tests.

Partial bias (PVB) occurs due to selective validation. Inverse probability bootstrap (IPB) is recommended for PVB correction. IPB showed higher standard error than other correction methods. For this situation, two methods, SIPW and SIPW-B, were suggested in the study. These gave better results than Sn and Sp.

The study was evaluated as a statistical method, and it was concluded that the proposed methods made significant contributions to the scientific method.

In the manuscript, especially in the discussion section, the importance of the statistical contribution of the proposed methods in terms of scientific research was not emphasized enough. In addition, it was observed that the literature support supporting the findings was not sufficient in the discussion and that recent studies were not cited as references.

The elimination of these deficiencies will bring the manuscript to a higher quality level.

Good luck.

Reviewer #3: Recommendations for Improving the Manuscript for PONE-D-25-12005

To ensure that the manuscript meets the editorial and ethical standards of PLOS ONE, the following revisions are recommended:

-Please include a clear and specific ethics statement in the Methods section.

-Although the study is based on simulations and secondary data, PLOS ONE requires a formal ethics declaration.

-In accordance with the PLOS ONE data availability policy, authors must provide unrestricted access to the data and code underlying the findings.

-The simulation setup should be described in sufficient detail to allow full reproducibility.

-While the results show improvements over existing methods, the Discussion section should explicitly acknowledge potential limitations.

6. PLOS authors have the option to publish the peer review history of their article (what does this mean?). If published, this will include your full peer review and any attached files.

Reviewer #1: No

Reviewer #2: No

Reviewer #3: No

---

## [Author Response · Author response to Decision Letter 1]

2 Sep 2025

Response to reviewers

PONE-D-25-12005

Partial Verification Bias Correction Using Scaled Inverse Probability Resampling for Binary Diagnostic Tests

Dear reviewers,

Thank you for the constructive comments given to the manuscript. We provide all responses to your comments below (and also in the "Response to reviewers.docx" attached with the submission):

Reviewer 1:

-------------

> 1-The summary should be expanded a bit.

This has been expanded to 300 words

> 2- The last sentence in the summary is not understandable, it should be rewritten

This sentence has been rewritten as “Although the methods are computationally demanding, ...”

> 3-The spelling of the “universiti” in the address section should be corrected.

> 9-The spelling of the word “universiti” in line 339 should be corrected.

This is the official name and spelling of the university (in Malay), which is indexed in academic databases. This should not be changed.

> 4-The sentence starting with "in the study" on line 34 and the rest should be written in a new paragraph.

“In the study” refers to Arifin and Yusof’s study (preceding sentence). Therefore, the paragraph starting with “In the study” has been moved to a new paragraph and “In the study” has been rephrased as “In their study...”.

> 5-The word "ın words" in line 62 should be corrected to "In the other words".

This has been corrected as suggested.

> 6-The sentence starting with "In addition" in line 171 is not understood.

The sentence starting from “In addition…” has been rephrased for clarity as suggested.

> 7-"The side effect" in line 207 this term is used in the medical field. Please write the sentence with another word.

This phrase has been replaced with “trade-off”.

> 8- A word expressing possibility should be used instead of "should" in line 328.

This has been replaced with “is expected to”.

> 10- Article spelling and grammar rules should be reviewed.

The comments related to the choice of words and sentence construction have been addressed accordingly. In addition, the article has been carefully proofread again as suggested.

Reviewer 2:

-------------

> The study was evaluated as a statistical method, and it was concluded that the proposed methods made significant contributions to the scientific method.

> In the manuscript, especially in the discussion section, the importance of the statistical contribution of the proposed methods in terms of scientific research was not emphasized enough. In addition, it was observed that the literature support supporting the findings was not sufficient in the discussion and that recent studies were not cited as references.

The discussion has been revised according to the comment. Discussion has been arranged in a way to better highlight the findings, and to facilitate comparison with literature. Relevant recent papers have been cited accordingly.

In addition, for clarity and to better reflect the flow of text in the discussion, some parts of the results section have been rearranged.

Reviewer 3:

-------------

> To ensure that the manuscript meets the editorial and ethical standards of PLOS ONE, the following revisions are recommended:

> -Please include a clear and specific ethics statement in the Methods section.

> -Although the study is based on simulations and secondary data, PLOS ONE requires a formal ethics declaration.

A new section “Ethics” has been added as required, covering the ethical aspect of simulation and secondary data analyses.

> - In accordance with the PLOS ONE data availability policy, authors must provide unrestricted access to the data and code underlying the findings.

This information is provided in PlosOne online submission form at Data availability section with a statement:

“The code and datasets used in this study are available at the following GitHub repository: https://github.com/wnarifin/sipw_in_pvb.”

We have also selected Yes to the question: Do the authors confirm that all data underlying the findings described in their manuscript are fully available without restriction?

In addition, the following texts have been added to experimental setup subsection:

“The code to generate and analyze the simulated data sets is available at https://github.com/wnarifin/sipw_in_pvb.”

“The clinical data sets and code to reproduce the results are available at https://github.com/wnarifin/sipw_in_pvb.”

> -The simulation setup should be described in sufficient detail to allow full reproducibility.

The “experimental setup” section has been expanded into two subsections: simulation setup and analysis, clinical data analysis.

The names of the subsections in Results have also been revised to “simulated data analysis” and “clinical data analysis” to reflect these changes.

> -While the results show improvements over existing methods, the Discussion section should explicitly acknowledge potential limitations.

The limitations are better highlighted in the Discussion section in the revised manuscript:

“While this study has demonstrated the strengths …” to “will be multiplied by 1000.” (the last paragraph in the discussion section)

----

In addition, we have made some changes to the affiliation of authors.

---

## [Editor Report · Decision Letter 1]

8 Sep 2025

Partial Verification Bias Correction Using Scaled Inverse Probability Resampling for Binary Diagnostic Tests

PONE-D-25-12005R1

Dear Dr. Arifin,

We’re pleased to inform you that your manuscript has been judged scientifically suitable for publication and will be formally accepted for publication once it meets all outstanding technical requirements.

Kind regards,

Hayrunnisa Nadaroglu

Academic Editor

PLOS ONE
---

## [Editor Report · Acceptance letter]

PONE-D-25-12005R1

PLOS ONE

Dear Dr. Arifin,

I'm pleased to inform you that your manuscript has been deemed suitable for publication in PLOS ONE. Congratulations! Your manuscript is now being handed over to our production team.

Kind regards,

on behalf of

Professor Hayrunnisa Nadaroglu

Academic Editor

PLOS ONE